# Implantation of a Dual-Chamber Automatic Cardioverter Defibrillator in a Patient with Persistent Left Superior Vena Cava: Case Report and Brief Literature Review

**DOI:** 10.3390/diagnostics10121071

**Published:** 2020-12-10

**Authors:** Mihai Cristian Haba, Andreea Maria Ursaru, Antoniu Octavian Petriș, Ștefan Eduard Popescu, Nicolae Dan Tesloianu

**Affiliations:** 1“Grigore.T. Popa” University of Medicine and Pharmacy, Iași 700115, Romania; cristi.haba@gmail.com (M.C.H.); antoniu.petris@yahoo.ro (A.O.P.); 2Department of Cardiology, Emergency Clinical Hospital “Sf. Spiridon”, Iași 700111, Romania; popescu.stefaneduard@yahoo.ro (S.E.P.); dan_tesloianu@yahoo.com (N.D.T.)

**Keywords:** persistent left superior vena cava, innominate vein, cardioverter defibrillator, prevention of sudden cardiac death

## Abstract

Persistence of the left superior vena cava (PLSVC) is a congenital anomaly reported in 0.3–0.5% of patients. Due to the multiple and complex anatomical variations, transvenous lead placement can become challenging. We report the case of a 47-year-old patient diagnosed with non-ischemic dilated cardiomyopathy with reduced left ventricular ejection fraction (LVEF—27%), who was referred to our clinic for implantation of a dual-chamber cardioverter defibrillator for primary prevention of sudden cardiac death. During the procedure we encountered an abnormal guidewire trajectory and after venographic examination we established the diagnosis of persistent left superior vena cava. After difficult implantation of a 7F defibrillation lead through the coronary sinus, we managed to place the atrial lead through a narrow brachiocephalic vein into the right atrial appendage. In this paper, we aim to illustrate the medical and technical implications of implanting a cardioverter defibrillator in patients with PLSVC, highlighting the benefit of identifying and utilizing both the innominate vein, and the left superior vena cava and coronary sinus for placement of multiple leads, which would otherwise have been impossible.

## 1. Introduction

The persistence of left superior vena cava (PLSVC) is a congenital vascular anomaly, characterized by the absence of the regression of the left superior vena cava (SVC). The prevalence of PLSVC in the general population is around 0.3–0.5% [1,2], and it can be even higher (1.3–9%) in the case of patients with other cardiac congenital anomalies [3]. Usually, PLSVC does not present clinical manifestations [4], therefore, the diagnostic is made incidentally during an imagistic evaluation (echocardiography, computed tomography or magnetic resonance angiography) or during procedures requiring the catheterization of the left subclavian vein. As a result, the anatomical changes of PLSVC become relevant, especially when patients require implantation of cardiac pacemakers and cardioverter defibrillators. In these situations, placing the leads through the coronary sinus (CS) into the right heart chambers becomes challenging, and it also poses problems for the stability of the leads and the function of the device. We report a case of dual chamber cardioverter defibrillator implantation for primary prevention of sudden cardiac death in a patient with dilated cardiomyopathy and PLSVC, together with the medical and technical difficulties we encountered.

## 2. Case Report

We report the case of a 47-year-old patient, previously diagnosed with non-ischemic dilated cardiomyopathy (DCM), who was referred to our clinic for the implantation of a transvenous implantable cardioverter-defibrillator (ICD). The patient did not present any family history of cardiovascular disease or cardiovascular risk factors. At admission, the patient was under treatment with angiotensin-converting enzyme inhibitor (ACEI), aldosterone antagonist and low dose beta blocker.

Physical examination revealed discrete peripheral edema, bradycardia—heart rate of 50 beats per minute (bpm) frequent premature beats and an arterial blood pressure of 120/60 mmHg. The electrocardiogram revealed sinus rhythm, 58 bpm, normal QRS axis, amputated R waves in V1–V4, notched R-wave, resembling the letter “M” in V5–V6, with an QRS complex of 110 milliseconds and polymorphic premature ventricular complexes (Figure 1).

The transthoracic echocardiography showed a dilated left ventricle—left ventricle end-diastolic diameter of 71 mm, with a reduced left ventricle ejection fraction (LVEF) of 27% (Figure 2) through diffuse hypokinesia of the left ventricular walls (Appendix A), moderate functional mitral regurgitation (Appendix A) and mild secondary pulmonary hypertension. The coronary sinus was slightly dilated, with a proximal diameter of 13 mm. Considering the diagnosis of DCM with a LVEF ≤ 35%, despite more than 3 months of optimal medical therapy, an ICD was indicated in order to reduce the risk of sudden cardiac death (class of recommendation I, level of evidence B) [5]. Due to high estimated percentage of pacing, we chose to implant a dual chamber ICD.

After standard left subclavian vein approach the guidewire presented an unexpected course descending on the left side of the spine. Therefore, we performed a contrast injection through the antero-cubital vein, which confirmed the presence of PLSVC, and also the presence of a very small caliber innominate (brachiocephalic) vein connecting the LSVC and the right superior vena cava (RSVC). We started implanting the 9F single coil defibrillation lead (Medtronic Spring Quattro^TM^, Model No. 6935M, length 62 cm, MR Conditional) through the LSVC and the CS. We encountered important resistance as we advanced the lead from the LSVC into the CS, due to a stenosis at the transition zone, the CS resembling a funnel-shaped expansion at the LSVC orifice. We chose to switch the 9F lead for a 7F single coil defibrillation lead (St. Jude Durata^TM^, Model No. 7170Q, length 65 cm), which eased the passage to the right atrium. After reaching the atrium, we used a J-shaped stylet to pass the lead through the tricuspid valve. After a few maneuvers, we placed the lead at the right ventricular apex, using active fixation. The lead parameters were satisfactory—pacing threshold 0.75 V/0.4 ms, R-wave sensing 8 mV, lead impedance 878 Ω and a shock impedance of 60 Ω. Afterwards, we tried to advance the atrial lead (Medtronic CapSureFix Novus^TM^, Model No. 5076, length 52 cm, MR Conditional) through the LSVC and the CS, but it affected the stability of the right ventricular lead. In this situation, we took advantage of the innominate vein: initially, we encountered some resistance during the advancement of the guidewire, with the impossibility to access the bridging innominate vein, the trajectory of the guidewire resembling that of the ventricular lead; after multiple attempts we approached the innominate vein with a 7-French sheath with the dilator within, thus, managing the access into the vein, and passage, initially of the wire, and then of the atrial lead across the bridging brachiocephalic vein into the RSVC and further into the right atrium. Finally, we placed the lead in the right atrial appendage (RAA), using active fixation, with satisfactory parameters—pacing threshold 0.5 V/0.4 ms, P-wave sensing 2.4 mV, lead impedance 677 Ω. We preferred advancing the atrial lead through the innominate vein and RSVC, in order to achieve a stabile position in the anterior RAA, since placing it through the CS implied fixating the lead on the free atrial wall, with increased risk of lead dislodgement and perforation. The final X ray, revealing the leads placement can be seen in Figure 3, Appendix A. 

After the procedure, the patient was discharged with increased dose of beta blocker, ACEI and aldosterone antagonist. One month and 6 month follow-ups of the device showed adequate defibrillator function, a few episodes of non-sustained ventricular tachycardia, with no need of anti-tachycardia pacing (ATP) or internal shock delivery. Routine echocardiography pointed out a much more dilated CS (Figure 4) when comparing to the initial echocardiography. A control venography confirmed the significant dilation of the CS 6 months postprocedural and, supplementary, revealed the obliteration of the innominate vein with no flow within (Figure 5, Appendix A). The localized stenosis at the border between LSVC and the CS, although not as significant as intraprocedural, is still visible (Figure 5b). We consider the increased venous flow, due to the postprocedural obliteration of the innominate vein, led to dilation of the stenosis and of the CS.

Subsequent computed tomography angiography excluded other venous or cardiac anomalies (Figure 6).

This paper was written in accordance with the Declaration of Helsinki of 1975, which was revised in 2013. The patient gave verbal and written informed consent and fully authorized the authors to use his medical data for research purposes, as stated in the attached journal written informed consent and in the “patient informed consent” approved by the Hospital Ethics Committee (protocol according to the Order 1410/12 December 2016, issued by the Romanian Ministry of Health), both signed by the patient.

## 3. Discussions

### 3.1. Epidemiology

Reports regarding PLSVC have been published since the 17th and 18th century, however, the first comprehensive description of this vascular malformation was presented by John Marshall in the 19th century [6]. Since then, various case reports and reviews have been published on the topic of PLSVC, especially after improvements of imagistic techniques and development of thoracic vein catheterization. PLSVC is the most common congenital anomaly of the thoracic venous system, with a prevalence under 0.5%, according to imagistic [7] and autopsy studies [8]. However, as PLSVC is mostly asymptomatic, the actual prevalence of this anomaly might be underestimated. In association with congenital heart disease (CHD), the incidence is even higher, reaching 9 to 10% [3,9]. Reports regarding CHD and PLSVC vary, due to demographics and the clinical characteristics of the studied population. According to a recent study that enrolled a total of 2.663 patients with CHD who underwent cardiac catheterization and angiography, of whom 88 (3.3%) were diagnosed with PLSVC, the most common associated acyanotic CHDs are atrial septal defect and persistent ductus arteriosus, while the most common associated cyanotic CHDs are ventricular septal defect, double outlet right ventricle and tetralogy of Fallot [3].

### 3.2. Embryology and Anatomical Variants

PLSVC is a result of anomalies in the embryological development of the thoracic venous system. In the first weeks of gestation, the embryonic venous system consists of two pairs of cardinal veins: two anterior cardinal veins, draining venous blood from the cranial half of the embryo and two posterior cardinal veins, draining the caudal half. These two pair of veins are collected into the right and left common cardinal veins which drain into the heart. In the eighth week of gestation, the oblique anastomosis between the two anterior cardinal veins begins to enlarge developing into the left brachiocephalic (innominate) vein. The right anterior cardinal vein goes on and develops into the right SVC. The part of the left anterior cardinal vein situated below the innominate vein obliterates, forming “the ligament of Marshall”. When the left anterior cardinal vein does not regress, it develops into the left SVC and it drains the venous blood into the coronary sinus. In case of other development defects, regarding the integration of the venous sinus into the right atrium, PLSVC can have different endings, such as the left atrium or pulmonary veins. In certain situations, the right anterior cardinal vein can be obliterated or the left brachiocephalic vein can be absent or rudimentary [2,10].

Schummer et al., proposed a classification presenting three anatomical types for the superior caval venous system. Type I corresponds to normal venous anatomy. Type II corresponds to PLSVC and the absence of the right superior vena cava. Isolated PLSVC without the presence of the right superior vena cava is of a very rare occurrence, with only 0.1% incidence rate in visceroatrial situs solitus [11,12], but increases to 40% with abnormal situs [13]. Type III is characterized by the presence of both superior vena cava. In 60% of these cases, the brachiocephalic vein is present (type IIIa), while in the others, it is absent (type IIIb) [10]. In more than 92% of situations, PLSVC drains venous blood into the right atrium through the coronary sinus [14]. However, in a small number of cases, the PLSVC can open directly into the left atrium or pulmonary veins, resulting in a right-left shunt, requiring surgical repair [15,16,17]. Other anatomical and embryological classifications have been published [18,19], but we consider that Schummer’s classification to have very good applicability in clinical practice. According to intraprocedural venography and CT angiography, our patient presented both SVC connected by a small caliber brachiocephalic (innominate) vein (type IIIa).

### 3.3. Clinical and Technical Implications

In the vast majority of cases, PLSVC is asymptomatic as the venous blood from LSVC is being directed into the right atrium through the CS, without hemodynamical changes in blood flow. The sinus node and the atrio-ventricular (AV) node are located during ontogenesis close to the cardinal veins and the sinus venosus, and they can be affected by the lack of regression of the LSVC. Therefore, patients with PLSVC can have a poorly formed sinus or AV node, which can be a substrate for sinus node dysfunction (SND), AV conduction disorders, arrhythmias or sudden cardiac death [4]. In the case of our patient, the dominating symptoms were those of heart failure in the context of DCM. However, the bradycardia secondary to low dose beta blockers could be an early sign of SND, prompting us to choose a dual-chamber cardioverter defibrillator.

PLSVC should be suspected after echocardiographic identification of a dilated CS (>1 cm). In this situation, an easy diagnosis test for PLSVC is performing a bilateral “bubble-study”, with injection of agitated saline in peripheral veins of both arms. In a patient with normal anatomy, bubbles should be seen only in the right atrium, but in patients with PLSVC, bubbles can be seen also in the coronary sinus [2]. Further investigations, such as computer tomography (CT) and magnetic resonance (MR) angiography, can offer more insight regarding the exact anatomy and other associated congenital anomalies. In our case, the diagnosis of PLSVC type IIIa was made incidentally, as the guidewire advanced on the left side of the spine. Afterwards, we confirmed the diagnosis using venography during the implantation procedure and by a subsequent thoracic CT angiography.

The main concerns for physicians implanting a cardiac device in a patient with PLSVC are achieving proper lead placement and function, preventing lead dislodgement and minimizing radiation exposure. The coronary sinus opens in the right atrium just above the tricuspid valve, forming a very steep angle with the valve, and also the RAA. As a result, advancing the defibrillator lead into the right ventricle and placing it at the apex or on the septum can be technically difficult. The number of bends that the lead forms by taking this trajectory (subclavian vein-LSVC-coronary sinus-right ventricle) increases the risk of lead dislodgement [19]. Several authors have proposed shaping the stylet manually in a U or J-shape, or looping the lead into the atrium, in order to pass through the tricuspid valve [4,8]. It is preferable to use active fixation leads in the case of selective site pacing, in order to avoid lead dislodgement [4,8,20,21,22]. Therefore, in our patient, we chose atrial and ventricular screw-in leads.

The persistence of LSVC results in a dilated CS, which is usually able to accommodate a 9F defibrillation lead. Our patient had a patent innominate vein, which rerouted a part of the venous blood, the CS presenting slight dilatation; additionally, we encountered a resistance when passing the lead from the LSVC into the CS, which led to the impossibility of accommodating the 9F lead. As a result, we switched to a 7F defibrillation lead which advanced through the LSVC and CS easier, and we managed to place it at the right ventricular apex. In situations when manipulation of the lead through the CS is difficult, authors have suggested moving the implantation procedure on the right side if the RSVC is present [4,8,22]. Crossing on the right side implies venographic confirmation of RSVC, an additional puncture and increased risk of pneumothorax and increase postprocedural discomfort in right-handed patients. In case of cardioverter defibrillators, right side placement might reduce the efficiency of defibrillation therapies, requiring defibrillation threshold testing [23]. As a result, there are case reports in which right sided approach was used for lead placement, and the lead was connected to the left side defibrillator pocket through subcutaneous tunneling [4,20].

The presence of the innominate vein was an important advantage for us, as we could use it in order to place the right atrium lead, without affecting the stability of the RV lead. Placing the atrial lead through the coronary sinus implies fixating the lead on the free atrial wall, increasing risk of lead dislodgement and perforation [24,25]. We managed to place the atrial lead into the right atrial appendage with a proper lead function and no post-procedural complications. This approach, in which a lead is placed through the LSVC and the other through the innominate vein and the RSVC has been reported by few other authors [26]. Papers reported placing a dual coil defibrillation lead through the innominate vein, in order to preserve a proper position of the SVC coil, while the atrial lead was placed through the CS [27,28]. Furthermore, there are reports of using balloon venoplasty when a stenotic or small caliber innominate vein is present [28]. In cases were the patient requires a CRT/CRT-D the innominate vein can be used for the atrial and RV leads, while LSVC might be used for the placement of the left ventricle lead into the CS and its branches [29,30]. In the case of our patient, we found that the lead configuration that we used presented with good stability, proper lead function, and no complications at the 6 month follow up.

## 4. Conclusions

PLSVC is a rare vascular congenital malformation, usually asymptomatic, that can pose technical difficulties in the context of transvenous implantation of a cardiac device. The present case shows that implantation of a dual chamber defibrillator is technically feasible, as long as the operator is willing to adapt to the specific anatomy of the patient, and also highlights a potential benefit of identifying and utilizing both the innominate vein, and the LSVC and CS for placement of multiple leads. However, we need to emphasize the fact that physicians implanting intracardiac devices should be familiar with PLSVC anatomic variants, and with the steps and techniques used to achieve proper device placement.

## Figures and Tables

**Figure 1 diagnostics-10-01071-f001:**
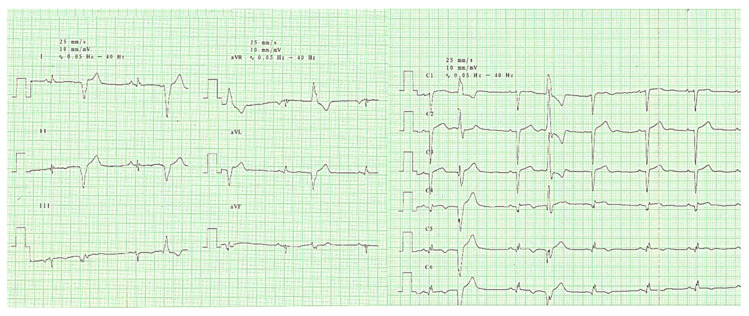
Electrocardiogram revealing sinus rhythm, 58 bpm, intermediate electric axis, amputated R waves in V1–V4, notched R-wave, resembling the letter “M” in V5–V6, with an QRS complex of 110 milliseconds and polymorphic premature ventricular complexes.

**Figure 2 diagnostics-10-01071-f002:**
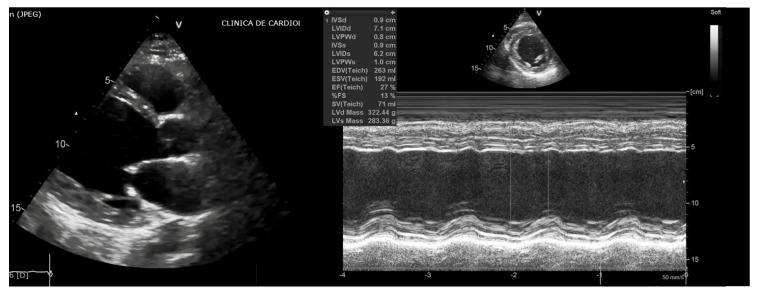
Two-dimensional echocardiography—parasternal long axis (**a**), parasternal short axis (**b**), revealing the increased diameter of the left ventricle (a left ventricular end diastolic diameter of 71 mm), with myocardial wall thinning and an ejection fraction of 27%.

**Figure 3 diagnostics-10-01071-f003:**
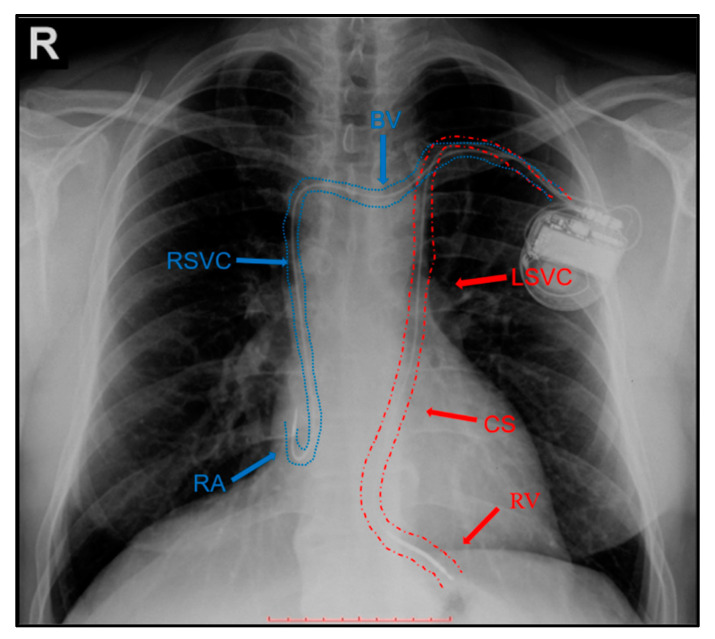
The defibrillation lead can be seen on the trajectory of the left superior vena cava (LSVC), through the coronary sinus (CS) into the right ventricular apex (RV)—red arrows. The atrial lead can be seen through the brachiocephalic vein (BV), right superior vena cava (RSVC) into the right atrial appendage (RA)—blue arrows.

**Figure 4 diagnostics-10-01071-f004:**
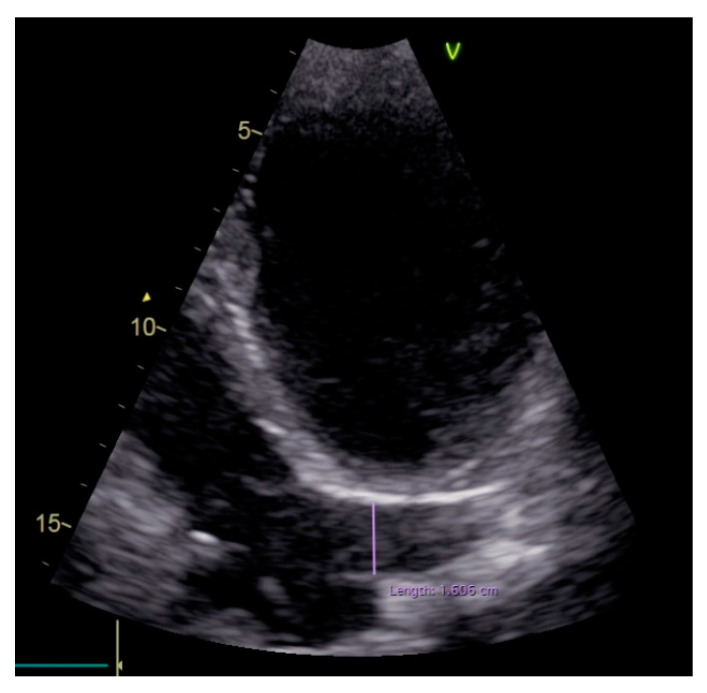
Two-dimensional echocardiography, apical four-chamber view showing a dilated CS (16 mm proximal).

**Figure 5 diagnostics-10-01071-f005:**
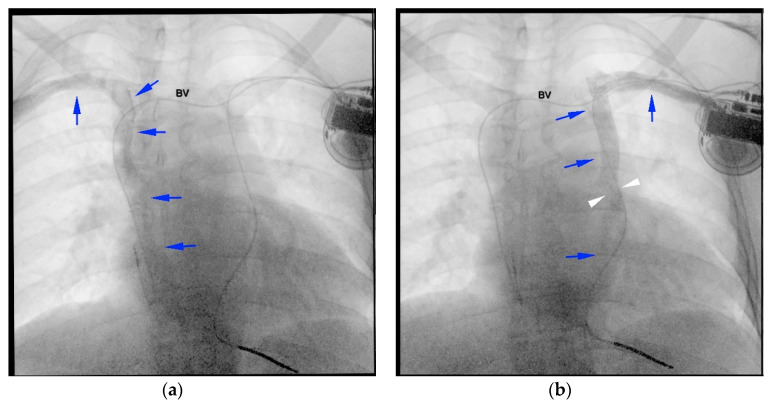
(**a**) Venography: right superior vena cava (SVC) draining into the right atrium (blue arrows); obstruction of the brachiocephalic vein (BV) with no contrast flow; (**b**) venography: left SVC draining in the right atrium via the coronary sinus (blue arrows); obstruction of the BV, with no contrast flow; residual stenosis/kinking at the border between left SVC and CS (white arrowheads).

**Figure 6 diagnostics-10-01071-f006:**
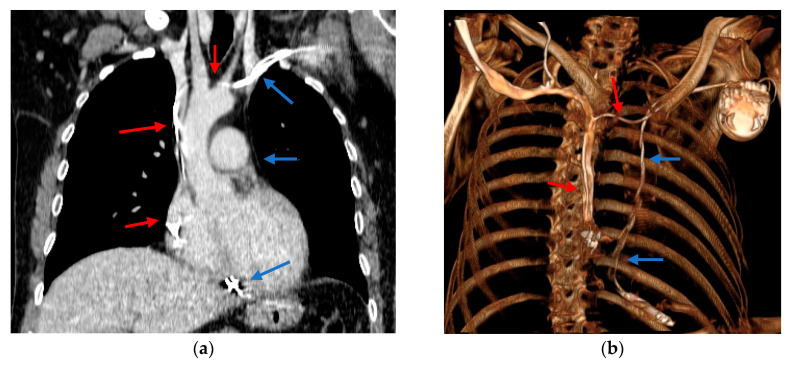
(**a**) Computed tomography angiography (venous phase) showing: the defibrillation lead going through the left SVC, the CS and, finally, into the RV (blue arrows); the atrial lead through the right SVC and into the RAA (red arrows); (**b**) 3D reconstruction of the CD system: defibrillation lead (blue arrows), atrial lead (red arrows;) Abbreviations: SVC—superior vena cava; CS—coronary sinus; RV—right ventricle; RAA—right atrial appendage; CD—cardioverter defibrillator.

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
