# Peer review of "Implantation of a Dual-Chamber Automatic Cardioverter Defibrillator in a Patient with Persistent Left Superior Vena Cava: Case Report and Brief Literature Review"

_diagnostics, 2020, doi:10.3390/diagnostics10121071_

Round 1
Reviewer 1 Report
The authors described the case of a 47-year-old patient diagnosed with nonischemic dilated cardiomyopathy with reduced left ventricular ejection fraction (LVEF 27%), who was referred to the clinic for implantation of a dual-chamber cardioverter defibrillator for primary prevention of sudden cardiac death. In this paper they aim to illustrate the medical and technical implications of ICD in patients with PLSVC, highlighting the benefit of identifying and utilizing both the innominate vein, and the left superior vena cava and coronary sinus for placement of multiple leads, which would otherwise have been impossible.
This case report is very interesting and well written.
Major comments
- This is a case where you are wondering whether to insert the atrial lead from CS or the brachiocephalic vein (BV). The size of CS and BV by venography before implantation may help determine the judgment. If possible, please show the venography. Why did you choose to insert the atrial lead from BV? Please comment.
- The CS looks slightly dilated in the long-axis image of the echo, is it correct?
- Figure of ECG was obscure, so please provide it clearly.
- Figure 3 showed the defibrillation lead on the trajectory of LSVC, through the CS into RV apex, which is partly unclear due to the silhouette of the heart and spine. Please make the defibrillation lead clear with dot lines.
Author Response
Dear Reviewer,
Thank you very much for your comments! We greatly appreciate it, they certainly allowed us to improve our paper. Beneath we tried to respond your comments.
Point 1. This is a case where you are wondering whether to insert the atrial lead from CS or the brachiocephalic vein (BV). The size of CS and BV by venography before implantation may help determine the judgment. If possible, please show the venography. Why did you choose to insert the atrial lead from BV? Please comment.
Response 1: Although the coronary sinus (CS) was slightly dilated, we encountered important resistance as we could not advance the 9F defibrillation lead from the left superior vena cava (LSVC) into the CS, due to a stenosis at the transition zone, and we chose to change it to a 7F lead. When trying to advance the atrial lead through the left LSVC and the CS, it affected the stability of the right ventricular lead. That is why, in the first place, we took advantage of the brachiocephalic vein (BV). Also, we preferred advancing the atrial lead through the BV, in order to achieve a stabile position in the anterior RAA, since placing it through the CS usually implies fixating the lead on the free atrial wall, with increased risk of lead dislodgement and perforation. Unfortunately, because of a system failure, the initial venography is not available anymore. We have attached (and made the comments in text), the 6 months venography, where the localized stenosis at the border between LSVC and the CS can still be seen, although not as significant as intraprocedural (Figure 5b, Video 5).
Point 2. The CS looks slightly dilated in the long-axis image of the echo, is it correct?
Response 2: We are sorry for the inadvertency, the CS was slightly dilated before procedure; we made the following change in line number 62: "was slightly dilated, with a proximal diameter of 13 mm."
It become even more dilated after the procedure; we have also attached the 6 months follow-up echocardiography (Figure 4).
Point 3. Figure of ECG was obscure, so please provide it clearly.
Response 3: We have provided a new scan for the ECG figure (Figure 1, line number 55).
Point 4. Figure 3 showed the defibrillation lead on the trajectory of LSVC, through the CS into RV apex, which is partly unclear due to the silhouette of the heart and spine. Please make the defibrillation lead clear with dot lines.
Response 4: We have modified Figure 3 (line number 95).
Also, starting from your comments, we added some supplementary figures, videos and text. The corrections/ additions are highlighted in red.
Reviewer 2 Report
Interesting case report about a patient with a patent LSVC requiring a cardioverter defibrillation.
The clinical case is interesting since it is a challenge that many cardiologists or cardiac surgeons have had to deal with. In general terms, I have no major comments, but I would like images or videos of the fluoroscopy.
Schummer et al classification: Type 2 PLSVC 18% of cases? I think it is too much
The writing is poor and there are many grammatical errors, for example (not only):
"he presented a class IB of implantation " This does not make sense
"The number..... increase"S
Tauras et al recommended (same verb tense throughout the entire paper)
Author Response
Dear Reviewer,
Thank you very much for your comments! We greatly appreciate it, they certainly allowed us to improve our paper. Beneath we tried to respond your comments.
Point 1: The clinical case is interesting since it is a challenge that many cardiologists or cardiac surgeons have had to deal with. In general terms, I have no major comments, but I would like images or videos of the fluoroscopy.
Response 1. We have provided videos of the fluoroscopy (Video 3 and Video 4).
Point 2: Schummer et al classification: Type 2 PLSVC 18% of cases? I think it is too much
Response 2. We are sorry for the inadvertency. We rephrased the sentence: “Isolated PLSVC without the presence of the right superior vena cava is of a very rare occurrence with only 0.1% incidence rate in visceroatrial situs solitus, [11,12] but increases to 40% with abnormal situs [13].” – line number 162; 3 supplementary references were necessary [11-13] (line number 288-292).
Point 3: The writing is poor and there are many grammatical errors, for example (not only):
"he presented a class IB of implantation " This does not make sense
"The number..... increase"S
Tauras et al recommended (same verb tense throughout the entire paper)
Response 3. We have rephrased the underlined grammatical errors (line numbers: 62-64, 197 and 222)
and tried to slightly improve our English writing throughout the manuscript. The corrections/ additions are highlighted in red.
Also, starting from your comments, we added some supplementary figures, videos and text. All corrections/ additions are highlighted in red.
Round 2
Reviewer 1 Report
There are no comments for your revision. Thank you for the change and the explanations.